# Mechanism of SOS PR-domain autoinhibition revealed by single-molecule assays on native protein from lysate

Young Kwang Lee[1,2], Shalini T. Low-Nam[1], Jean K. Chung[1], Scott D. Hansen[1], Hiu Yue Monatrice Lam[1], Steven Alvarez[1] & Jay T. Groves[1,2]

The guanine nucleotide exchange factor (GEF) Son of Sevenless (SOS) plays a critical role in signal transduction by activating Ras. Here we introduce a single-molecule assay in which individual SOS molecules are captured from raw cell lysate using Ras-functionalized supported membrane microarrays. This enables characterization of the full-length SOS protein, which has not previously been studied in reconstitution due to difficulties in purification. Our measurements on the full-length protein reveal a distinct role of the C-terminal proline-rich (PR) domain to obstruct the engagement of allosteric Ras independently of the well-known N-terminal domain autoinhibition. This inhibitory role of the PR domain limits Grb2-independent recruitment of SOS to the membrane through binding of Ras·GTP in the SOS allosteric binding site. More generally, this assay strategy enables characterization of the functional behaviour of GEFs with single-molecule precision but without the need for purification.

[1] Department of Chemistry, University of California, 408A Stanley Hall, Berkeley, California 94720, USA. [2] Molecular Biophysics and Integrated Bioimaging Division, Lawrence Berkeley National Laboratory, Berkeley, California 94720, USA. Correspondence and requests for materials should be addressed to J.T.G. (email: jtgroves@lbl.gov).

The guanine nucleotide exchange factor (GEF), Son of Sevenless (SOS), is a critical intermediary that transduces receptor tyrosine kinase stimulation into Ras activation[1–5]. SOS activity is also essential in T-cell receptor signalling, and, in this case, has been reported to act as an analogue-to-digital signal integrator[6–9]. The adaptor protein Grb2 mediates SOS recruitment to the membrane via docking to phosphorylated tyrosine residues on receptors or scaffold proteins[10–19]. Once at the membrane, lipid-binding domains as well as the Ras allosteric binding pocket on SOS engage their membrane ligands, locking SOS to the membrane where it can processively catalyse nucleotide exchange on thousands of Ras molecules[20,21]. Recent live-cell experiments confirm that once activated, SOS molecules can remain bound to the membrane and active until they are ultimately internalized via endocytosis[22]. Allosteric Ras binding has been identified as the key step to sustain SOS activity at the membrane, both in reconstitution and in live cells[21–24]. SOS1 mutations that impair autoinhibition and allosteric regulation result in constitutive Ras activation[25], and have been implicated in developmental disorders such as Noonan syndrome[25,26].

Regulation of SOS originates from the collective operations of its modular domains[27–30]. The catalytic core of SOS, SOS[Cat], accommodates two Ras molecules: one at the allosteric site spanning the REM and CDC25 domains, and the other at the catalytic site in the CDC25 domain (Fig. 1a). Non-substrate Ras binding at the allosteric site leads to a marked enhancement of the nucleotide exchange activity[1]. Ras·GTP is a more potent allosteric activator of SOS than Ras·GDP, thereby creating a positive feedback loop in which SOS is activated by its own product[8,31]. This nucleotide-specific allosteric activation is largely accomplished through changes in the kinetic rate of SOS recruitment to the membrane; individual SOS molecular catalytic rates do not change appreciably for truncated constructs[20,21] or the full-length protein (as shown here). SOS[Cat] is flanked by the C-terminal proline-rich (PR) domain and amino-terminal domains composed of two histone folds and Dbl-homology and Pleckstrin-homology domains (Fig. 1a). The N-terminal domains concertedly mask the allosteric site and prevent spurious SOS recruitment[32,33]. Interactions with anionic lipids, especially phosphatidylinositol 4,5-bisphosphate (PIP$_2$), relieve this autoinhibition[23,33,34].

In contrast to the detailed characterization of the N-terminal domains of SOS, the role of the carboxy-terminal PR domain has remained elusive. This is due to the intrinsically disordered structure of the PR domain[35], which complicates both purification and crystallization of functional, full-length SOS. The main role for the PR domain is the recruitment of SOS to activated receptors via binding to the SH3 domain of Grb2 (refs 36,37). There is evidence suggesting an additional inhibitory role for the PR domain, but mechanistic understanding of this important effect is lacking[22,29,38,39]. Truncated SOS constructs, lacking the C-terminal domain, are able to bypass Grb2-mediated membrane recruitment and act as potent Ras activators[39,40]. Similarly, mutations that cause a premature stop codon and abolish the PR domain promote oncogenic transformation[39] and have been found in hyperplastic syndromes such as hereditary gingival fibromatosis[41,42]. These observations lead to a model in which the PR domain directly affects accessibility to Ras and thus modulates the kinetic rate of allosteric activation and/or molecular catalytic rates. However, these hypotheses have not been directly addressed due to unavailability of the full-length SOS protein.

Here we adapt a recently introduced single-molecule SOS activity assay using Ras-functionalized supported lipid membrane microarrays to study full-length SOS protein captured from raw cell lysates. Membrane binding, molecular diffusion and catalytic rates from various SOS constructs, including the full-length protein, were analysed. These measurements reveal that the PR domain has a distinct function to obstruct allosteric Ras binding, independent of N-terminal autoinhibition. Complete autoinhibition of SOS requires both the N- and C-terminal inhibitory modes, which can be relived independently. Relief of autoinhibition strongly enhances the kinetic rates of membrane recruitment and allosteric activation. This inhibition provides critical control over the Grb2-independent activation of SOS via allosteric binding to Ras·GTP, which has emerged as the predominant mechanism of positive feedback in the activation of Ras by SOS. Using our assay strategy, we quantitatively measured both ectopic and endogenous GEF activity from cell lysate at the single-molecule level. This method could be broadly useful to characterize functional behaviours of individual proteins in membrane environments without the need for protein purification.

## Results

**Characterization of SOS expressed in HEK293T cells.** A series of SOS constructs, including the native, full-length protein were transiently expressed in HEK293T cells and collected in crude, whole-cell lysates (Fig. 1a). Several SOS constructs were fused with an enhanced green fluorescence protein (EGFP) tag at the C terminus to enable single-molecule visualization. SOS molecules in lysates were characterized by western blotting (Fig. 1b and Supplementary Fig. 1) and single-molecule counting analysis. Approximately 90% of each of full-length SOS (SOS[FL]) and SOS[HDPC] was expressed at the expected molecular weight (Fig. 1c). SOS[Cat] and SOS[CatPR] were expressed as pure species within the limit of detection. Typically, transfected SOS is expressed at least several fold excess over endogenous SOS in the cell lysate (Supplementary Fig. 1). The concentration of each EGFP-tagged SOS construct was calculated using a standard curve of purified EGFP fluorescence (Supplementary Fig. 2). Protein stoichiometry can be determined by measuring single fluorophore photobleaching steps[43,44]. Traces of fluorescence from single EGFP-tagged SOS[FL] molecules immobilized on a glass substrate exhibited predominantly single step photobleaching, consistent with the monomeric state (Fig. 1d,e). These results demonstrate that full-length SOS constructs are successfully expressed in HEK293T cells and are monomeric in lysates.

**Single-molecule SOS activity assays from lysate.** Here we adapt a recently developed membrane microarray assay platform that enables real-time observation of Ras activation by individual SOS molecules on the membrane[21]. The strategy is briefly described as follows: H-Ras (1-181, C118S) was tethered to the supported membrane via covalent crosslinking between the terminal cysteine (Cys[181]) and maleimide-functionalized lipids[21,45,46]. Ras was loaded with a fluorescent, non-hydrolysable analogue of guanosine triphosphate, Atto 488-labelled GppNp (henceforth referred to as GTP). Mobility and surface density of Ras were measured by fluorescence recovery after photobleaching and fluorescence correlation spectroscopy (FCS) (Supplementary Figs 3 and 4). Typical Ras lateral mobilities and densities in these experiments were $\sim 3.0\,\mu m^2\,s^{-1}$ and 300–1,200 Ras $\mu m^{-2}$, respectively. The membrane-tethered Ras and SOS were laterally confined in an array of micrometre-scale supported lipid bilayers (SLB) that are partitioned by nanofabricated chromium metal lines (Fig. 2a)[47–49]. In this assay, SOS constructs (without the EGFP tag) in cell lysate were incubated with the Ras-coupled bilayers. After rinsing free and loosely bound SOS molecules, the exchange reaction of SOS stably recruited to the membrane was initiated by flowing in free nucleotides. In each corral, a single SOS catalyses the exchange of fluorescent GTP-bound

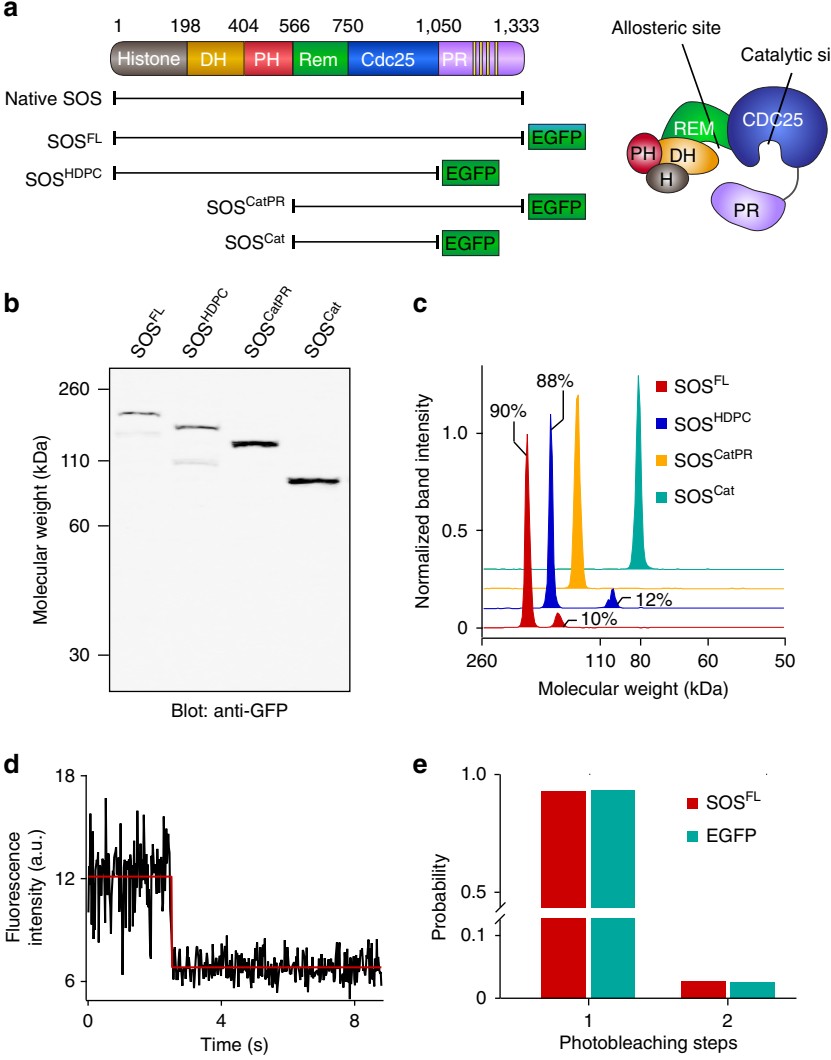

**Figure 1 | Characterization of various SOS constructs in HEK293T cell lysate.** (**a**) Domain organization of SOS. Native SOS and various EGFP-fused SOS constructs are shown. (**b**) Anti-GFP western blotting of SOS constructs from lysates. (**c**) Intensity profiles of western blot bands. Approximately 90% of $SOS^{FL}$ and $SOS^{HDPC}$ were expressed at the correct molecular weight. $SOS^{Cat}$ and $SOS^{CatPR}$ were expressed as pure species. (**d**) Representative single-step photobleaching trace of $SOS^{FL}$. (**e**) Histogram of photobleaching steps for $SOS^{FL}$ from lysate compared with purified, monomeric EGFP. $SOS^{FL}$ predominantly exists in a monomeric state. $N$ is $\sim 200$ for each protein.

Ras with unlabelled GTP in solution. Thus, the nucleotide exchange reaction was directly measured as a decrease in fluorescence in individual corrals (Fig. 2b and Supplementary Fig. 5). The concentration of SOS was maintained such that >95% of active corrals contained exactly one enzyme (see Methods for details)[21].

We measured individual enzyme activities from ectopically expressed SOS constructs, as well as endogenous GEFs, in HEK293T cell lysates (Fig. 2b). In all cases, GEFs are captured from lysate by their ability to stably associate with the Ras functionalized membrane and processively catalyse nucleotide exchange for many minutes. In untransfected cells, a few corrals successfully capture a GEF such as endogenous SOS, enabling direct analysis of these endogenously expressed proteins. The same concentration of the transfected cell lysate exhibited far more active corrals, indicating that the ectopically expressed SOS constructs greatly outnumbered endogenous GEFs.

**The PR domain does not alter the molecular catalytic rate.**
Native full-length SOS (see Supplementary Fig. 6a for the western

blotting) were highly processive on the membrane surface and individual molecules exhibited discrete transitions between well-defined catalytic states (Fig. 2c and Supplementary Fig. 7). We have previously reported this long timescale dynamic heterogeneity in truncated SOS constructs and proposed that such fluctuations may contribute a dynamic mechanism of allostery that emerges at the level of the signalling network[21]. These data on native SOS provide the first confirmation that the full-length protein exhibits the same long timescale dynamic activity fluctuations.

Next, we compared catalytic rate distributions of endogenous GEFs, full-length SOS and $SOS^{HDPC}$. The catalytic rates were estimated from individual functional substates of single-enzyme activity traces. The rate distribution of native full-length SOS shows a high degree of overlap with endogenous GEFs in untransfected cell lysate, suggesting that the measured endogenous GEF are likely to be SOS molecules (Fig. 2d). However, other endogenous GEF molecules could also be detected if they exhibit sufficient processivity. Single-molecule enzyme kinetics of full-length native SOS was also similar to $SOS^{HDPC}$ in terms of the catalytic rate and state lifetime (Fig. 2d and Supplementary

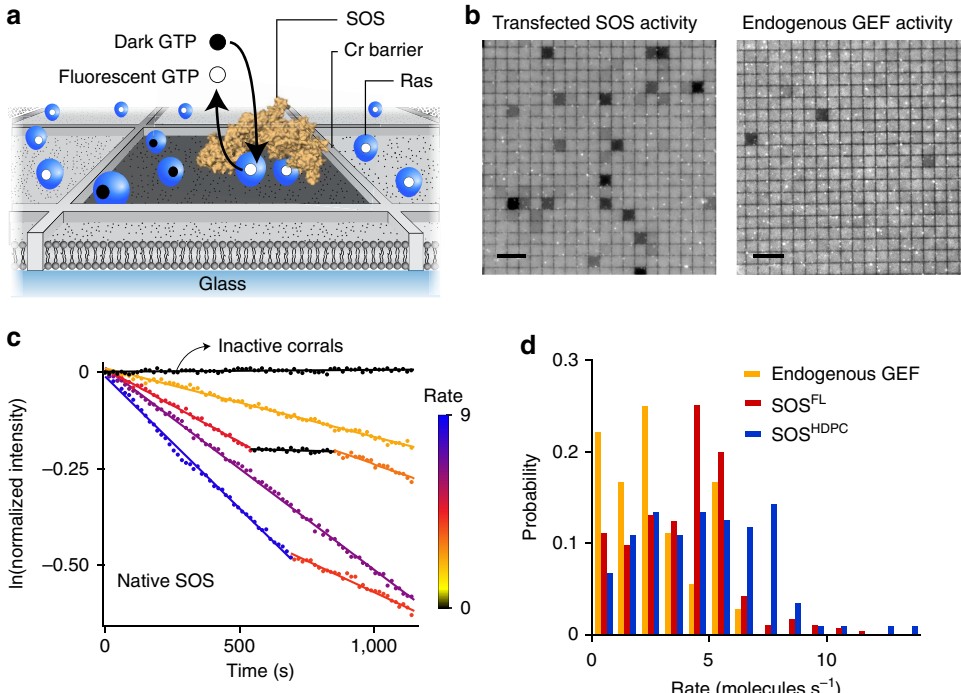

**Figure 2 | The PR domain does not significantly alter single-molecule nucleotide exchange kinetics of SOS.** (**a**) Scheme of the single-molecule SOS activity assay. Membrane-tethered Ras molecules, bound to fluorescent nucleotides, are laterally confined in an array of micrometer-scale supported lipid bilayers that are partitioned by nanofabricated chromium metal lines. Stably recruited SOS catalyses nucleotide exchange for excess non-fluorescent nucleotide present in solution. (**b**) Epifluorescence images of fluorescently loaded Ras on a partitioned bilayer, at the end of the observation. A subset of corrals underwent nucleotide exchange and became dark. Nucleotide exchange by SOS from transfected and untransfected cell lysates are shown on the left and right, respectively. (**c**) Single-enzyme kinetic traces for native SOS. Discrete activity states are identified as linear segments with distinct slopes. Each enzymatic state is colour-coded according to its catalytic rate (molecules s$^{-1}$). (**d**) Catalytic rate probability distributions of endogenous GEF (yellow; average and s.d. of rates: $2.79 \pm 1.76$ molecules s$^{-1}$), native full-length SOS (red; $3.89 \pm 1.98$ molecules s$^{-1}$) and SOS$^{HDPC}$ (blue; $4.68 \pm 2.65$ molecules s$^{-1}$). N of active corrals: ~150 for each SOS$^{FL}$ and SOS$^{HDPC}$, ~80 for endogenous GEF. Lipid composition (in mol%): egg-PC/MCC-DOPE/DOPS = 94/3/3. Surface density of Ras: ~800 μm$^{-2}$. Scale bars, 10 μm.

Fig. 8). The majority of SOS$^{HDPC}$ rates overlap with native SOS, with some faster rates. The similar enzymatic kinetics of native SOS and SOS$^{HDPC}$ indicate that any inhibitory effects of the PR domain are largely suppressed by stable membrane engagement. We have not observed a marked enhancement of catalytic rate of individual SOS molecules when both the N- and C-terminal regulatory modules are fully removed, that is, SOS$^{Cat}$ (ref. 21). Thus, we find no evidence that SOS regulation involves significant changes in the molecular catalytic rate on the membrane, suggesting that the autoinhibitory domains regulate the enzyme activity through another process in SOS activation.

**Steady-state nucleotide exchange assay.** To determine the effects of the PR domain on the activation kinetics of SOS, we performed steady-state nucleotide exchange assays on the membrane microarrays. In this assay configuration, diluted lysate was maintained in the reaction chamber at a constant concentration, to measure the nucleotide exchange of Ras, while SOS undergoes activation processes such as membrane recruitment and allosteric activation. This assay configuration contrasts the single-molecule activity assays (mentioned above and in ref. 21), in which SOS molecules that have already activated and become stably associated with the membrane are measured. We performed the steady-state assays with GTP-loaded Ras to promote allosteric activation of SOS. As each corral could interact with multiple SOS molecules, the ensemble SOS activity determines the nucleotide exchange profile of individual corrals. Catalytic nucleotide

exchange was measured by quantifying reduction in surface fluorescence of Ras and corrected for the contribution of intrinsic Ras turnover and photobleaching (Supplementary Fig. 9). We confirmed that other components of the lysate have minimal effects on exchange activity across a range of concentrations (Supplementary Fig. 10).

**The PR domain suppresses SOS activation on membranes.** Steady-state reaction measurements in the presence of SOS in solution reveal that the PR domain suppresses activation of SOS on the membrane. Full-length SOS (SOS$^{FL}$) had significantly lower activity compared with the truncated constructs with only a small fraction of corrals turning dark (Fig. 3a–c). The probability distribution of catalytic nucleotide exchange sampled from thousands of corrals is plotted (Fig. 3d). For SOS$^{FL}$, most of the corrals show a minimal level of nucleotide exchange and fall under a Gaussian distribution. This result indicates that the majority of SOS$^{FL}$ is highly autoinhibited. The rare dark corrals represent a second population, distinctly outside the major Gaussian population, and are broadly distributed over a range of high nucleotide exchange values (indicated by arrows in Fig. 3d). This rare population is attributed to the presence of single, highly processive SOS molecules that achieve stable allosteric activation on the membrane (for example, the molecular activation state studied in the activity assays)[21].

We propose that these highly processive, long-lived SOS molecules may be disproportionately significant in the context of

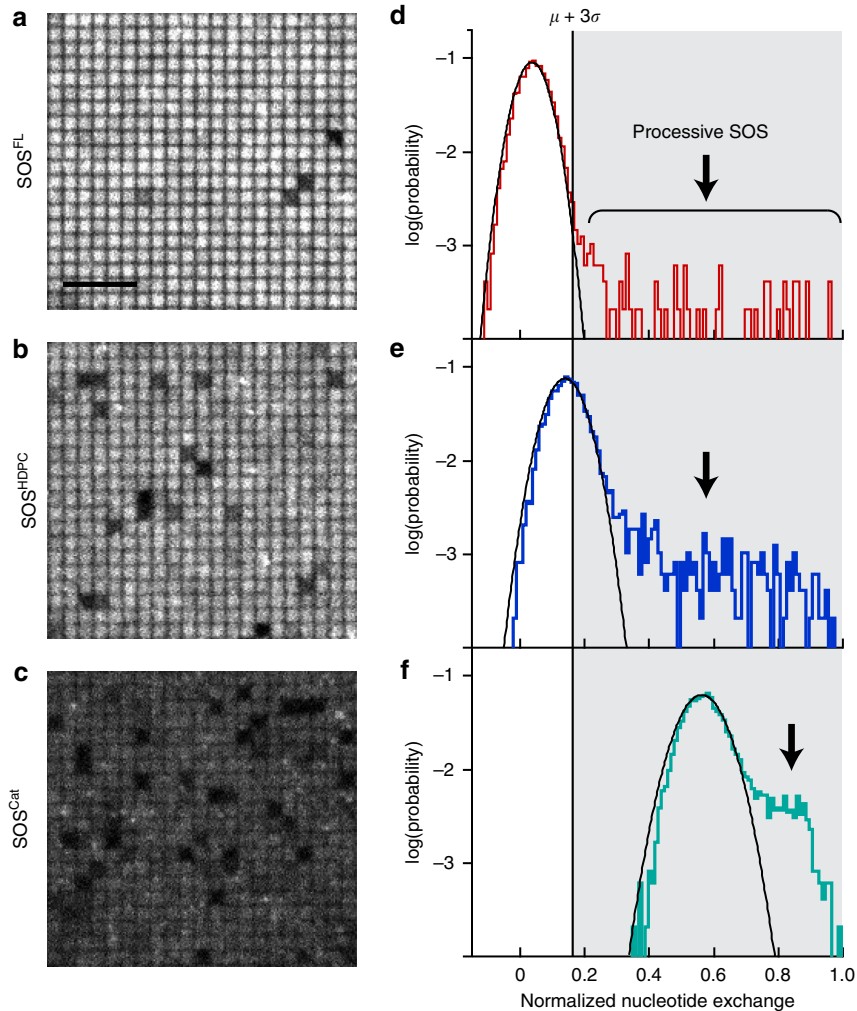

**Figure 3 | The PR domain suppresses activation of SOS.** Fluorescent nucleotide-loaded Ras on a partitioned bilayer was constantly exposed to 1 nM SOS for 10 min. Epifluorescence images of Ras-functionalized bilayer catalysed by (**a**) SOS$^{FL}$, (**b**) SOS$^{HDPC}$ and (**c**) SOS$^{Cat}$. Normalized nucleotide exchange probability distributions of (**d**) SOS$^{FL}$, (**e**) SOS$^{HDPC}$ and (**f**) SOS$^{Cat}$. The distributions are acquired from surface fluorescence intensities of 4,800 corrals for each SOS construct. The major populations, showing weak nucleotide exchange, are adequately fitted by Gaussian distributions. The second, minor populations correspond to high nucleotide exchange driven by highly processive SOS. The hypothetical activation threshold ($\mu + 3\sigma$) is set between these two populations for SOS$^{FL}$. Lipid composition (in mol%): egg-PC/MCC-DOPE/DOPS = 94/3/3. Surface density of Ras: $\sim 280\,\mu m^{-2}$. Scale bar, 5 μm.

cell signalling[21,22]. They not only activate many Ras molecules, but Ras are all activated in nearly the same place and within a short time interval. This concentrated burst of Ras activation may be sufficient to overcome the constitutive negative pressure of Ras GTPase-activating proteins, which deactivate Ras by enhancing the rate of GTP hydrolysis to GDP[4,21]. Under such a mechanism, the Ras activation signal that could lead to downstream activation of mitogen-activated protein kinase and other pathways would predominantly come from these rare, highly processive SOS molecules. The more dynamic activation by Ras from transiently membrane associated SOS would represent an inconsequential background. In this context, we analysed the effects of N- and C-terminal regulatory domains by examining a hypothetical activation threshold of the mean plus 3 s.d. ($\mu + 3\sigma$) of the major, minimally active population for full-length SOS (Fig. 3d). The threshold delineates the inactive and active nucleotide exchange regimes. It is worth noting that this hypothetical threshold could be different from the cellular activation threshold, which involves more complex signalling networks.

SOS$^{HDPC}$ showed a significant increase in the occurrence of the highly active corrals that overcame the activation threshold via processive enzyme activity (the dark corrals in Fig. 3b and the non-Gaussian second population in Fig. 3e). Given the high degree of similarity in molecular catalytic rates between SOS$^{FL}$ and SOS$^{HDPC}$ (Fig. 2d), the enhanced activity is probably due to a higher activation probability for SOS$^{HDPC}$ on the membranes. In addition, truncation of the PR domain gives rise to weak nonspecific activity. Some corrals in the major Gaussian population overcame the activation threshold without processive SOS activity (Fig. 3e). It is likely to be that, in the absence of the PR domain autoinhibition, brief membrane encounters by multiple SOS enzymes from solution are responsible for this background rate of nucleotide exchange. The enhancement of catalysis by truncation of the PR domain is consistent with allosteric effects. Both forms of enhanced catalytic activity (weak nonspecific activity and processive activity) in SOS$^{HDPC}$ were diminished with Ras·GDP (Supplementary Fig. 11). This nucleotide state sensitivity is the hallmark of allosteric regulation

of SOS—the allosteric site has a higher affinity to Ras·GTP than Ras·GDP[31].

As expected, the truncation of both N- and C-terminal regulatory modules further enhances the kinetic rate of SOS activation, shifting the entire population of the histogram beyond the hypothetical activation threshold (Fig. 3f). Remarkably, the nucleotide exchange ability of SOS[Cat] from solution (the Gaussian-fitted population) reached the level of membrane-bound, processive SOS[FL], emphasizing autoinhibition as the key mechanism to prevent spurious Ras activation. These results demonstrate that both N- and C-terminal domains prevent activation of SOS and downregulate its activity.

**The PR domain regulation is independent of the N terminus**. Above, we showed that the PR domain suppresses SOS activity by reducing the kinetic rate of activation. This activation occurs exclusively on the membrane surface, implicating membrane recruitment as an important regulatory step for SOS activation[22]. We therefore measured binding rates of the various EGFP-tagged SOS constructs to Ras-modified SLBs to quantitatively characterize the inhibitory contributions of the N- and C-terminal domains (Fig. 4a). EGFP-tagged SOS that dwell on the membrane surface were selectively visualized using a total internal reflection microscopy at the single-molecule level. We confirmed that interaction with Ras is required for SOS to reside on the membrane longer than the time resolution of our measurements (43 ms). SOS showed a very low level of detectable binding on a Ras-free bilayer (Supplementary Fig. 12). To determine the effects of lipid interactions on SOS binding to Ras, the cumulative binding of each of the SOS constructs were measured on membranes of various compositions: egg-PC membranes doped with 3 mol% of either 1,2-dioleoyl-sn-glycero-3-phospho-L-serine (DOPS, henceforth referred to as PS) or PIP$_2$ (Fig. 4b). The linear trends in the cumulative binding traces show that SOS binds to the membrane at constant rates. The extracted slope of the cumulative binding trace is the binding frequency (Fig. 4c,d) and the molar binding rate was estimated from the slope of a linear fit of the binding frequency obtained with various SOS concentrations (Fig. 4e).

It has been shown that the N-terminal domains occlude the allosteric Ras binding site and inhibit SOS activation[33]. The inhibitory effect is relieved by interactions with various membrane lipids such as PIP$_2$ (refs 23,33,34,50). We observed that this classical regulatory mechanism of the N-terminal domains operates in the full-length enzyme similar to truncated constructs. With the truncated constructs, SOS[HDPC] and SOS[Cat], we confirmed that appending the full N-terminal domains has a noticeable damping effect on Ras binding (Fig. 4c). PIP$_2$ almost fully relieved the N-terminal inhibitory effect in SOS[HDPC] and significantly increased its binding kinetics to Ras (Fig. 4c). The direct binding of SOS to PIP$_2$ in the absence of Ras was undetectable, because the timescale of this interaction is faster than the time resolution of our single-molecule binding assays (Supplementary Fig. 13). Nevertheless, the transient interaction of PIP$_2$ is sufficient to induce a productive conformational rearrangement of N-terminal domains. We also observed membrane-dependent regulatory behaviours of the N-terminal domains in the full-length enzyme by comparing SOS[FL] with SOS[CatPR], which is full-length SOS lacking the three consecutive N-terminal domains. SOS[FL] exhibited lower Ras binding compared with SOS[CatPR] and this inhibitory effect was also almost eliminated in the presence of PIP$_2$ (Fig. 4d). This indicates that the PR domain does not alter the inhibition by the N-terminal domains or their responsiveness to the signalling lipids.

A comparison of molar binding rates reveals a negative regulatory function of the PR domain through weakening the affinity to Ras. SOS[CatPR] showed a lower binding rate compared to SOS[Cat] (Fig. 4e). This indicates that the PR domain has its own inhibitory effect, regardless of the N-terminal domains. In full-length SOS, the PR domain cooperates with N-terminal domains to achieve the complete autoinhibition. SOS[FL] exhibits the additional reduction in recruitment to the Ras-functionalized PS bilayer compared with SOS[HDPC] (Fig. 4e). Importantly, PIP$_2$ specifically mitigated autoinhibition of the N-terminal domains without affecting that of the PR domain. SOS[FL] was still autoinhibited on the PIP$_2$ bilayer through the PR domain, showing a similar binding rate to SOS[CatPR] (Fig. 4e). These results demonstrate that the regulatory mechanisms of N- and C-terminal modules are independent of each other and full autoinhibition requires both the N- and C-terminal inhibitory modes. The observed membrane-dependent binding behaviours show good agreement with SOS activity in steady-state nucleotide exchange assays, confirming that the membrane recruitment is a critical regulatory mechanism of SOS (Supplementary Fig. 14).

**The PR domain does not influence the N-terminal lipid sensing**. We characterized the lipid interactions with SOS by analysing step size distribution of single-molecule trajectories[51–53]. This further supports that the marked changes in the binding rate on PIP$_2$ bilayers are associated with specific interactions with the N-terminal domains and these lipid interactions are not affected by the PR domain. To examine how N- and C-terminal domains interact with lipids, various SOS constructs were tracked on Ras-functionalized SLBs containing either PS or PIP$_2$ (see Methods for more details).

The step size distribution for SOS[Cat] trajectories is adequately described with a two-species Brownian diffusion model (Fig. 5a). The two diffusing species probably correspond to two different binding states of SOS[Cat]—bound with either one or two Ras molecules. Ras behaves strictly as a single species at the typical surface density (Supplementary Fig. 15). Diffusive motions of all other SOS constructs agree with two-species model. SOS[Cat] and SOS[CatPR] show nearly identical diffusion on the PS and PIP$_2$ bilayer (Fig. 5a,b). In contrast, we observed an N-terminal domain-dependent mobility shift of SOS with different bilayer compositions. SOS[HDPC] interacts with PIP$_2$ on the membrane surface, resulting in slower mobility on the PIP$_2$ bilayer compared with the PS bilayer (Fig. 5c). SOS[FL] also shows the similar trend (Fig. 5d). These results indicate that the PR domain does not impact the ability of the N-terminal domains to sense signalling lipids and explain why PIP$_2$ selectively relieves autoinhibition by the N-terminal domains in the full-length protein, in agreement with the binding rate measurements (Fig. 4d,e).

**The PR domain blocks allosteric Ras binding**. We demonstrated that the PR domain has the ability to independently inhibit Ras binding to SOS, suggesting that it blocks the access of Ras to either the allosteric or catalytic site. To assess the effect on the catalytic site, the allosteric site was mutationally disrupted using SOS[FL(W729E)] and SOS[HDPC(W729E)] (see Supplementary Fig. 6b for the western blot analysis of these two constructs)[32]. For both SOS[FL] and SOS[HDPC], binding of Ras to the catalytic site was abolished when the allosteric site binding was disrupted with the W729E mutation (Fig. 6a). This is consistent with prior observations that the catalytic site of SOS is virtually inactive without allosterically bound Ras[31,32], and suggests that allosteric Ras engagement initiates membrane recruitment of SOS when bypassing Grb2-mediated translocation. We still cannot rule out an interaction of the PR domain with the catalytic site.

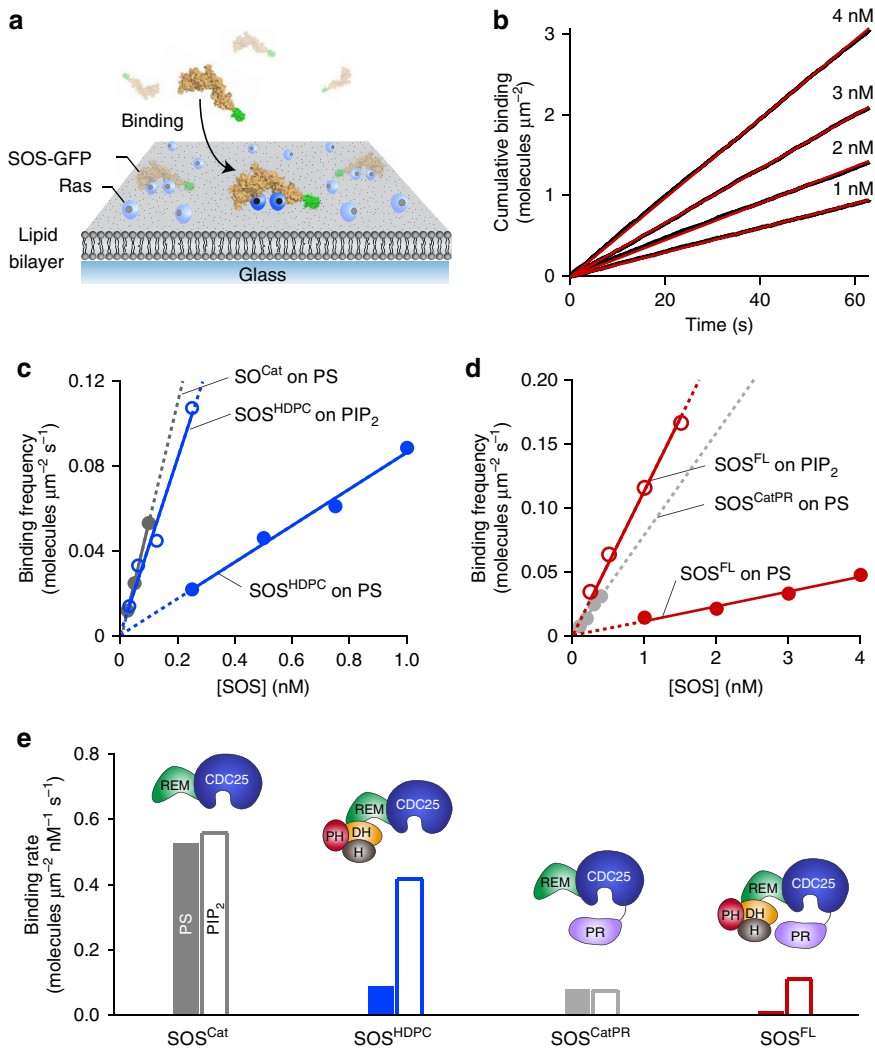

**Figure 4 | The PR domain regulates Ras binding to SOS independently of the N-terminal domains.** (**a**) Scheme of SOS binding assay on a Ras-modified supported lipid bilayer. When SOS in lysate is introduced with free nucleotides in solution, SOS engages both allosteric and catalytic Ras, and catalyses nucleotide exchange on the membrane. SOS molecules dwelling on the membrane surface are selectively visualized using TIRF microscopy. (**b**) Representative cumulative binding curves. The cumulative binding of SOS$^{FL}$ to Ras on a PS bilayer as a function of time (red lines). Linear fits of cumulative binding curves are shown as black lines; the slope of these fits represents the concentration-dependent binding frequency. (**c,d**) The concentration-dependent binding frequency of various SOS constructs on the Ras-modified lipid membrane containing either 1,2-dioleoyl-sn-glycero-3-phospho-L-serine (DOPS) (solid circles) or PIP$_2$ (empty circles). Lines represent extrapolated linear fits; the slope represents the binding rate. (**e**) The binding rate of various SOS constructs. Linear fits were applied with the y-intercept set to zero. Lipid composition (in mol%): egg-PC/MCC-DOPE/DOPS or PIP$_2$ = 94/3/3. Surface density of Ras: ∼1,200 μm$^{-2}$.

However, the low affinity of the catalytic site without allosterically bound Ras suggests that the interaction between the PR domain and catalytic site is a minor contribution to overall regulation of membrane recruitment.

Next, we examined the inhibitory effect of the PR domain on Ras engagement at the allosteric site utilizing a mutant form of Ras, Ras$^{(Y64A)}$, which exclusively engages the allosteric site of SOS[54]. SOS$^{Cat}$ and SOS$^{CatPR}$ constructs were observed to remove inhibitory contributions from the N-terminal domains. In comparison with SOS$^{Cat}$, SOS$^{CatPR}$ showed a significant reduction in binding to Ras$^{(Y64A)}$, suggesting that the PR domain directly obstructs Ras binding to the allosteric site (Fig. 6b). The magnitude of reduction in SOS$^{CatPR}$ binding observed with Ras$^{Y64A}$ (∼5.0-fold) was similar to that with wild-type Ras (∼6.7 fold) shown previously (Supplementary Table 1). This further supports that the PR domain prevents membrane recruitment of SOS by blocking allosteric Ras binding.

This raises the question how does SOS overcome the autoinhibition imposed by the PR domain? We tested the interaction with its well-known binding partner, Grb2, which binds the PR domain and translocates SOS to activated receptors[10–17]. Autoinhibition was not relieved by pre-incubation with Grb2 in solution, prior to exposure to Ras (Fig. 6c). This is not necessarily surprising, because SOS constantly interact with Grb2 in the cytoplasm and stays in an autoinhibited state[29,40]. It is likely to be that membrane-specific interactions with Grb2 are required to fully relieve autoinhibition and activate SOS.

## Discussion
It has been shown that the catalytic core of SOS, SOS$^{Cat}$, can bypass Grb2-mediated membrane recruitment through allosteric Ras binding and processively activate thousands of Ras

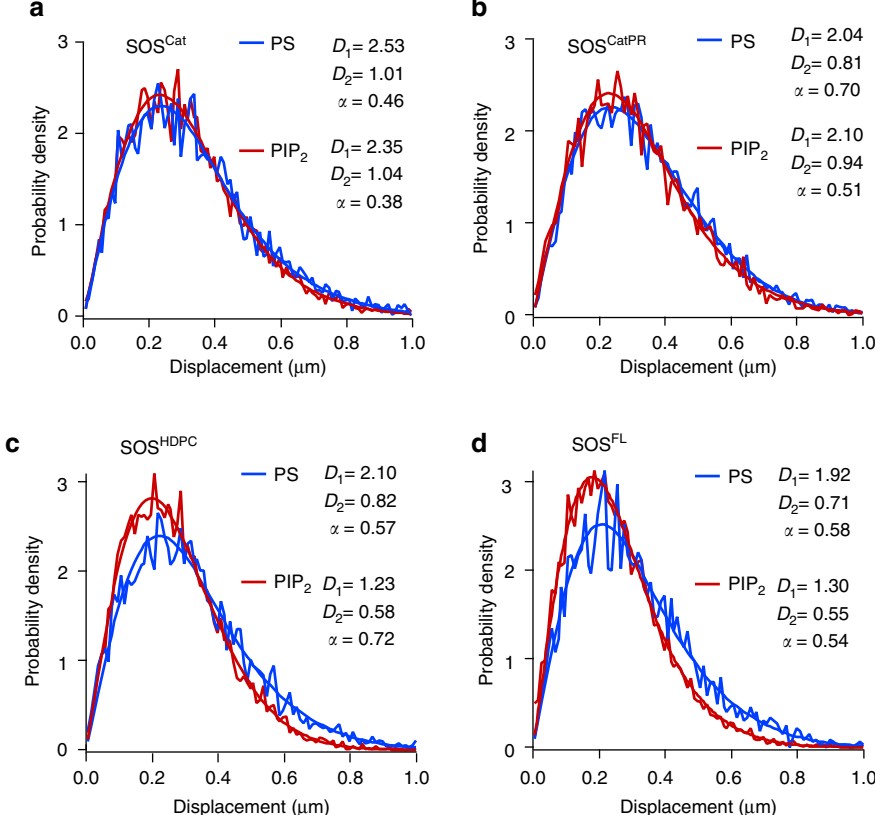

**Figure 5 | Step size distribution analysis reveals that the PR domain does not interfere with lipid interactions with N-terminal domains.** The step size distribution was acquired from multiple single molecule diffusion trajectories for (**a**) SOS$^{Cat}$, (**b**) SOS$^{CatPR}$, (**c**) SOS$^{HDPC}$, (**d**) SOS$^{FL}$ on the Ras-modified lipid membrane containing either 3 mol% of 1,2-dioleoyl-sn-glycero-3-phospho-L-serine (DOPS) or PIP$_2$. All data are adequately fit to a two-species diffusion model. The extracted coefficients were shown in the insets. $D_1$ and $D_2$ correspond to diffusion coefficients of the fast and slow mobility species (unit: $\mu m^2 s^{-1}$). $\alpha$ corresponds to the fraction of the fast species. The N-terminal domains slow SOS mobility on the PIP$_2$ bilayer, presumably due to specific interaction with the Pleckstrin-homology (PH) domain (**c**,**d**)[33]. Lipid composition (in mol%): egg-PC/MCC-DOPE/DOPS or PIP$_2$ = 94/3/3. Surface density of Ras: ~1,200 $\mu m^{-2}$.

molecules[21,23]. This could lead to erroneous allosteric activation of SOS and unbalance Ras-GTP levels, with devastating consequences for cells such as unchecked mitogenic signalling. SOS has a built-in molecular mechanism by which such uncontrolled activation is concertedly suppressed. The critical mechanism is inhibition of allosteric Ras engagement by the N-terminal regulatory modules. Higher affinity of the allosteric site for Ras·GTP over Ras·GDP also suppresses SOS activation in the basal state[31,33]. Spatiotemporal specificity is further conferred by multiple intermolecular interactions on the membrane such as with lipids and Grb2 that bind to the N- and C-terminal domains, respectively[28].

The data presented here add new mechanistic details to our understanding of allosteric regulation of SOS. The C-terminal PR domain has a distinct role in obstructing the engagement of allosteric Ras (Fig. 7). This PR domain autoinhibition is not suppressed by interactions with Grb2 in solution but largely relieved when SOS achieves stable allosteric Ras engagement on the membrane surfaces. The complementary roles of the PR domain in preventing allosteric Ras engagement and targeting the Grb2 SH3 domain essentially ensure receptor stimulation-specific SOS activation. Several known cancer-associated mutations in SOS1 are truncations in the PR domain, suggesting loss of PR domain autoinhibition may contribute to human cancer[22,55]. We have further shown that the PR domain autoinhibition is independent of N-terminal domains and the complete autoinhibition requires a collective operation of both N- and C-terminal inhibitory modes. Relief of either N- or C-terminal autoinhibition strongly enhances the kinetic rate of allosteric activation, which determines SOS activity. Thus, coincident detection of multiple input signals that relieve each inhibitory mode of SOS may maximize the efficiency of signal propagation in cells[28]. Together, our observations show how SOS coordinates its multidomain architecture on the membrane surface to regulate spatiotemporal specificity in signal transduction.

Finally, our strategy demonstrates single-molecule assays with crude cell lysate on a reconstituted membrane platform as a way to study full-length SOS, which is considered difficult to purify. This method can be applied to other proteins that are not available in a purified, functional form. The cell lysate-based single-molecule assays could reveal specific functional states of proteins under normal regulation and posttranslational modifications that occur in the cell[43,56–60].

## Methods

**SOS plasmids and H-Ras preparation.** Mammalian expression vector pEF6-human SOS1 (1-1333 aa; accession number: AK290228.1) were used for transient transfection of the native full-length protein[61]. EGFP-tagged SOS plasmids were created in the same expression vector with coding sequence for full-length SOS (SOS$^{FL}$, 1-1333 aa), SOS$^{HDPC}$ (1-1049 aa), SOS$^{Cat}$ (566-1049 aa) and SOS$^{CatPR}$ (566-1333 aa). EGFP coding sequence was added after SOS coding sequences, to yield the C-terminal tag. H-Ras$^{C118S}$ (H-Ras construct containing residues 1–181 with a single cysteine at position C181, henceforth referred to as Ras) and H-Ras$^{Y64A,C118S}$ (henceforth referred to as Y64A Ras) were cloned into a pProExHTb vector (Invitrogen). Ras was expressed in *Escherichia coli* and purified using an N-terminal

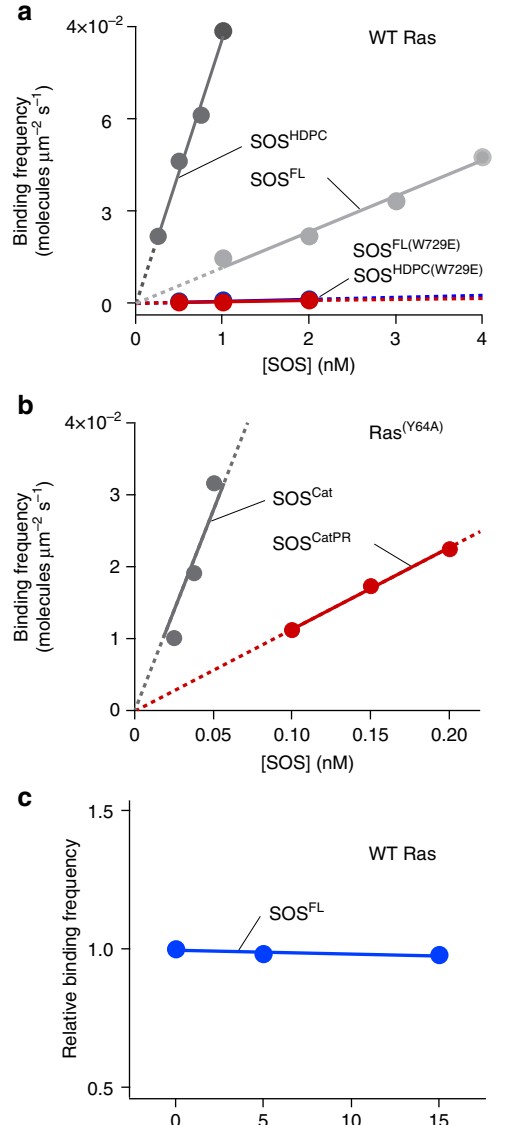

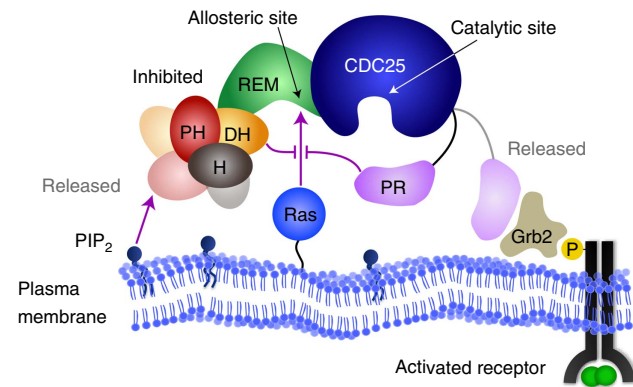

**Figure 7 | The complete autoinhibition of SOS requires allosteric inhibitory modes of both the N- and C-terminal regulatory modules.** The regulatory modules control kinetic rates of membrane recruitment and allosteric activation of SOS. The inhibitory functions of N- and C-terminal regulatory modules are independent of each other. Distinctive membrane interactions such as with lipids or activated receptors may release each inhibitory conformation and modulate allosteric activation probability of SOS in cells.

plasmids was incubated with 9 μg of polyethylenimine in reduced serum medium (Opti-MEM, Gibco) for 10 min to form DNA/polymer complexes and then introduced to cells in DMEM medium. After 5 h of expression, cells were washed with fresh medium and collected in lysis buffer containing Tris-HCl pH 7.4, 136 mM NaCl, 1% protease inhibitor cocktails (P8340, Sigma), 0.5% phosphatase inhibitor cocktails 2 (P0044, Sigma) and 3 (P5726, Sigma), 15 μg ml$^{-1}$ benzamidine, 50 mM sodium fluoride and 0.1 mM phenylmethanesulfonyl fluoride. After lysing cells by tip sonication on ice, the cell lysate was centrifuged at 20,000 g for 30 min at 4 °C. The supernatant was collected and used for single-molecule experiments. The fluorescence intensity of recombinant EGFP was measured using a fluorescence spectrometer (Carry Eclipse, Varian) and used to determine the concentration of EGFP-tagged SOS in cell lysate. For western blotting, prepared lysates were run on SDS–PAGE gels and transferred to PVDF membranes (Immobilon-P, Millipore). The membranes were incubated with either mouse anti-SOS1 (610095, BD Biosciences, 1:500 dilution) or mouse anti-GFP (sc-9996, Santa Cruz Biotechnology, 1:500 dilution). Subsequently, membranes were incubated with IRDye 800-conjugated goat-anti-mouse (926-32210, LiCor, 1:1,000 dilution) and visualized using the Odyssey Infrared Imaging System (LiCor). The western blotting band intensities were analysed using ImageJ software.

**Optical microscopy.** Epifluorescence and total internal reflection fluorescence (TIRF) images were acquired using a Nikon Eclipse Ti inverted microscope equipped with a × 100 1.49 numerical aperture oil-immersion TIRF objective and an Andor iXon electron-multiplying charge-coupled device camera. A mercury arc lamp was used for epifluorescence illumination. Lasers (488 and 637 nm; Coherent, Inc.) were used as illumination sources for TIRF imaging. ET500LP and ET525/50M filters (Chroma Technology Corp.) were used for 488 nm channel imaging. ET660LP and ET700/75M filters were used for 637 nm channel imaging.

**Fluorescence correlation spectroscopy.** Surface density and mobility of Ras were quantitatively analysed using FCS. Dual-colour FCS was performed on a home-built setup based on a Nikon Eclipse TE2000-E inverted microscope[46]. Briefly, excitation wavelengths were selected by bandpass filters from a pulsed white light laser source (SuperK Extreme EXW-12, NKT Photonics) and combined through a single-mode optical fibre. The excitation pulses enter the microscope via a multi-colour dichroic cube (Di01-R405/488/561/635-25x36, Semrock). The fluorescence signal is collected by a × 100 high-numerical aperture oil-immersion objective and recorded by avalanche photodiode detectors (Hamamatsu). The signal is directly converted into autocorrelation signal by a hardware correlator (Correlator.com). Lights (488 and 568 nm) were used to excite the Atto 488 fluorophore and Texas Red-DHPE, respectively. The resulting autocorrelation $G(\tau)$ was fit to the two-dimensional Gaussian diffusion model to calculate surface density and mobility of Ras[46].

**Ras-functionalized SLB preparation.** SLBs were prepared on plain or chromium corral-patterned glass coverslips. Chromium patterns were fabricated by e-beam lithography[62]. Glass substrates were cleaned with 2% Hallmanex III solution (Hellma Analytics) for 30 min followed by 15 min bath sonication in IPA/H$_2$O 1:1 and 3 min piranha etching in H$_2$SO$_4$/H$_2$O$_2$ 3:1. Substrates were rinsed with copious amounts of ultrapure water after each cleaning procedure. SLBs were prepared on cleaned glass

**Figure 6 | Inhibition of allosteric Ras binding to SOS by the PR domain.** (**a**) SOS$^{FL(W729E)}$ and SOS$^{HDPC(W729E)}$, SOS mutants that are impaired in allosteric Ras binding, show a very low level of membrane binding compared with wild-type SOS. (**b**) The allosteric site of SOS$^{CatPR}$ is autoinhibited by the PR domain and shows a lower binding affinity to Ras$^{(Y64A)}$ than SOS$^{Cat}$. Ras$^{(Y64A)}$ exclusively binds to the allosteric site. Linear fits were applied with the y intercept to be zero. (**c**) Relative binding frequency of 1 nM SOS$^{FL}$ at different concentrations of recombinant Grb2. Lipid composition (in mol%): egg-PC/MCC-DOPE/DOPS = 94/3/3. Surface density of Ras: ~1,200 μm$^{-2}$.

hexahistidine tag[23]. Following elution, the hexahistidine tag was cleaved by treating the tobacco etch virus protease overnight at 4 °C, while dialysing in 50 mM PBS (pH 8.0), 300 mM NaCl and 0.5 mM 2-mercaptoethanol. Ras was further purified by size-exclusion chromatography on a Superdex 75 column (GE Healthcare) that was equilibrated in gel-filtration buffer (20 mM Tris-HCl pH 8.0, 200 mM NaCl, 10% glycerol (v/v) and 1 mM (2-carboxyethyl)phosphine).

**Cell transfection and lysate preparation.** HEK293T cells were cultured in DMEM medium, high glucose media (DMEM, Gibco) supplemented with 10% fetal bovine serum, minimal essential medium (MEM) non-essential amino acid and 100 μg ml$^{-1}$ penicillin and streptomycin. Transfection was performed with linear polyethylenimine (Polysciences) as a carrier material. Typically, 3 μg of

coverslips using the vesicle fusion method[63,64]. Briefly, chloroform mixtures of 94 mol% L-α-phosphatidylcholine (Egg-PC, Chicken), 3 mol% 1,2-dioleoyl-sn-glycero-3-phosphoethanolamine-N-[4-(p-maleimidomethyl)cyclohexanecarboxamide] (MCC-DOPE) and either 3 mol% of 1,2-dioleoyl-sn-glycero-3-phospho-L-serine (DOPS) or L-α-phosphatidylinositol-4,5-bisphosphate (PIP₂, Porcine brain) were dried for 20 min at 25 °C. Dried lipid films were resuspended in PBS (pH 7.4) by vortexing. Small unilamellar vesicles were formed by extruding 17 times through a 30 nm polycarbonate filter (EMD Millipore) using a mini-extruder (Avestin). SLB formation and experiments were performed in flow chambers (sticky-Slide VI[0.4], Ibidi) assembled with prepared glass substrates. Glass substrates were first incubated with small unilamellar vesicle solution (0.5 mg ml$^{-1}$ in PBS) for 30 min to form SLB. Next, SLBs were incubated with casein (1 mg ml$^{-1}$ in PBS) for 10 min and then with Ras (typically 0.5 mg ml$^{-1}$ in PBS) for 2.5 h. Unreacted 1,2-dioleoyl-sn-glycero-3-phosphoethanolamine-N-[4-(p-maleimidomethyl)cyclohexanecarboxamide] was quenched by 10 min incubation with 2-mercaptoethanol (5 mM in PBS). Native nucleotides in H-Ras were stripped by 20 min EDTA incubation (50 mM in loading buffer consisting of 40 mM HEPES, 150 mM NaCl at pH 7.4) at 4 °C. SLBs were washed with loading buffer and incubated overnight with 10 μM nucleotides in reaction buffer (40 mM HEPES, 100 mM NaCl, 5 mM MgCl₂ at pH 7.4). Atto 488-labelled guanosine diphosphate (EDA-GDP-Atto 488), Atto 488-labelled guanosine triphosphate non-hydrolysable analogue (EDA-GppNp-Atto 488) and GppNp were purchased from Jena Bioscience. Next day, Ras-functionalized bilayers were brought to room temperature right before use and washed with 2 ml reaction buffer to remove unbound nucleotides.

**Single-molecule SOS activity assays on membrane microarrays.** In single-molecule corral experiments, Ras was functionalized on patterned SLBs and loaded with EDA-GppNp-Atto 488. Diluted cell lysate in reaction buffer was injected as a pulse through the reaction chamber with a flow rate of 0.5 ml min$^{-1}$ and then unbound SOS molecules were washed out. Lysate concentrations were adjusted so that ∼10% or less of all corrals in the array are enzymatically active (turn dark). In this situation, >95% of active corrals is catalysed by a single enzyme[21]. SOS-mediated nucleotide exchange reactions were initiated by providing a stable flow (0.5 ml min$^{-1}$ for first 2 min and 0.1 ml min$^{-1}$ afterwards) of 120 μM unlabelled GTP in reaction buffer together with 10 mM beta-mercaptoethanol (BME) and 0.1 mg ml$^{-1}$ casein. Time-lapse epifluorescence images were collected to measure the reaction kinetics for up to 20 min. Detailed analysis methods of single-molecule corral experiments have been described in Supplementary Fig. 5 (ref. 21). Briefly, time-dependent average intensity traces for individual corrals were extracted using Matlab. Each kinetic trace was normalized to its own initial maximum intensity value and then corrected for intensity changes due to photobleaching and intrinsic Ras turnover by dividing with the average trace from all corrals without SOS activity. Distinct kinetic states sampled by individual SOS enzymes were quantified using the change point detection algorithm (Supplementary Fig. 5)[21]. The rate constant for fluorescence decay per SOS, $k_{sos}$, was calculated by fitting individual kinetic traces obtained from distinct functional substates of SOS to the following equation: $\ln(I_{sos}(t)) = -k_{sos} \times t$. $I_{sos}(t)$ represents fluorescence decay by catalytic nucleotide exchange (see Supplementary Fig. 5). $k_{sos}$ was converted to enzymatic rates (TN) by the following equation: TN = $k_{sos} \times$ Ras(0). Ras(0) represents the number of Ras bound with fluorescence nucleotides at time 0.

**Steady-state nucleotide exchange on membrane microarrays.** Cell lysates were diluted to yield desired EGFP-SOS concentrations (typically 1 nM) in reaction buffer supplemented with 320 μg ml$^{-1}$ glucose oxidase (Serva Electrophoresis GmbH), 50 μg ml$^{-1}$ catalase (Sigma), 2 mM trolox, 20 mM beta-mercaptoethanol (BME), 20 mM glucose and 0.2 mg ml$^{-1}$ casein. SOS concentration was adjusted based on fluorescence intensity calibration standards of recombinant EGFPs. Lysates with 120 μM unlabelled GDP or GTP were introduced over SLBs functionalized with fluorescent nucleotide-loaded Ras. Lysates were maintained in the reaction chamber throughout the measurements. Time-lapse epifluorescence images were recorded. Intensity traces for individual corrals were extracted using ImageJ and Matlab. Each kinetic trace was normalized to its own initial maximum intensity value. Intensity changes due to enzymatic nucleotide exchange reactions were calculated by dividing with the average trace of inactive corrals obtained with the same concentration of untransfected lysate (Supplementary Fig. 9). For this, total protein concentrations in transfected and untransfected cell lysates were determined by Bradford assay (Bio-Rad) and diluted accordingly.

**Single-molecule binding and step size distribution analysis.** EGFP-SOS constructs were incubated with SLBs functionalized with unlabelled GppNp-loaded Ras. Binding assays were performed in the presence of 120 μM unlabelled GTP in supplemented reaction buffer, which is described in the above section. For single-molecule tracking experiments, SOS was incubated in the absence of free nucleotides in solution such that SOS becomes trapped by Ras at the catalytic site following nucleotide release. Binding and diffusion of EGFP-SOS molecules on SLBs was measured using a TIRF microscopy setup. Individual binding events and diffusion trajectories were analysed with TrackMate (ImageJ plugins) and Igor Pro (WaveMetrics). The step size distribution for each sample was fit with the

Brownian diffusion model as described by the following equation:

$$p(r, t, D) = \frac{ar}{2D_1 t} \exp\left(-\frac{r^2}{4D_1 t}\right) + \frac{(1-\alpha)r}{2D_2 t} \exp\left(-\frac{r^2}{4D_2 t}\right)$$

$D_1$ and $D_2$ are different diffusion coefficients for fast and slow species, respectively. $\alpha$ is a relative population for the fast species. For single-species diffusion model, $\alpha$ is 1, thus eliminating the second term. Fitting residues for the single- and two-component model were monitored to determine the number of diffusion species. The diffusion coefficients and relative population of each component were calculated from the corresponding fitting.

**Code availability.** The code for associated with this paper is available from the corresponding author on request.

**Data availability.** The authors declare that the data supporting the findings of this study are available within the paper and its Supplementary Information files or from the corresponding author upon request.

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

## Acknowledgements
This work is supported by NCI U01CA202241.

## Author contributions
J.T.G. supervised the project. J.T.G. and Y.K.L. conceptualized and designed the experiments. Y.K.L. and S.T.L.-N. performed experiments. Y.K.L. analysed data. S.D.H. assisted with DNA cloning. J.T.G., Y.K.L., S.T.L.-N. and J.K.C. wrote the paper. All authors discussed and commented on the results.

## Additional information

**Competing interests:** The authors declare no competing financial interests.

**Publisher's note**: 

