## [Peer Review File · Nature Communications]

Reviewers' Comments:

Reviewer #1 (Remarks to the Author)

The manuscript by Lee et al is an extension of the Groves' laboratory's previous work assessing SOS-dependent Ras activation at the single molecule level. They had previously shown that the N-terminal region of SOS inhibits membrane recruitment and downstream signaling. Significant to this manuscript, the authors show that the PR region of SOS also inhibits SOS membrane recruitment and therefore its activation. Further, they show that this is independent of the PIP2-dependent relief of autoinhibition provided by the N-terminal region of SOS.

Overall, the manuscript is of high quality and does help further our understanding of SOS signaling. However, the "mechanism of SOS PR-domain autoinhibition" is not fully explored. What seems to be missing is experimental evidence showing that the PR region's autoinhibition can be relieved physiologically. Presumably, this is due to Grb2 recruitment to the membrane, as Grb2 placed in solution was insufficient to relieve the autoinhibition. See major comment 1 below.

Major comments

1. The major goal of the manuscript is to show the mechanism of SOS PR-domain autoinhibition, and here the prevailing theory is that this autoinhibition is relieved by PR interactions with Grb2 at the membrane. The authors attempted (Fig. 6C) to enhance Ras/SOS binding by adding increasing amounts of Grb2 in solution, however, this did not relieve autoinhibition...the argument being that this needs to occur in the presence of membrane-specific Grb2 interactions. However, it would really bolster the paper and the conclusions if the authors could show this.

The Groves laboratory recently published a manuscript in PNAS where they were able to place phosphorylated LAT in their lipid bilayer, and then used this to show LAT:Grb2:SOS higher-order assemblies. Could the authors use a similar approach where both pLAT and RAS were in the membrane, and then repeat the experiment in Fig. 6C?

2. In Fig. 4, the authors use a nice, full set of truncation mutants to assess SOS/Ras binding on PS or PIP2 containing membranes, showing that the PR region blocks "most" of SOS recruitment/binding to Ras, and that the N-term region makes Ras binding PIP2-dependent. This

experiment is complete, well done, and is the best data in the manuscript.

In Fig 3 the authors show that removal of the PR region increases “catalytic activation” (or increases the kinetic rate of activation...I think that these are the same) on a PS bilayer, and that this is further increased by removal of the HPD domains. If this assay were instead done in the presence of PIP2, would SOS(Cat) and SOS(HDPC) show similar rates of activation?

Furthermore,

would a SOS(CatPR) construct show a similar level of allosteric activation to the SOS(HDPC) construct on PS, and would this only show differential activity if the assay were performed with PIP2 present?

Minor comments

1. The manuscript seems “maximally brief”. As it stands parts are read, as many of the complex assays/mechanisms are not explained in detail, making this hard to read for the molecular biology crowd. It would help if the authors would expand the manuscript to make it more broadly accessible. One example of this is on page 5, the authors refer to “state lifetime” and show data in Supplemental figure 7. It would be helpful if this term were explained somewhere (even if in the Supp 7 fig legend) to aid in understanding the manuscript.

2. In Fig 2D, the authors say that the catalytic rate of endogenous GEF, SOS, and SOS(HPCD) were all “similar”...however looking at the data it seems that there is a shift in the peak/distribution of these three populations. Further, similar changes in their earlier papers (see Science 2014 SOS(cat) vs SOS(HPCD)) were viewed as different. Is there some statistical way to analyze whether these are in fact “similar” or not on a population level?

3. On page 4 (bottom), when assessing exchange activity in untransfected cells, the authors say that this activity is “presumably SOS”, and allows for direct analysis of endogenous protein. While there is “unpublished data” from the Roose lab mentioned in their 2007 paper that RasGRP proteins are not expressed in 293T cells, do we know the same is true for RasGRF1/2? Also, would it be better to say that they are analyzing an endogenous activity since we are not positive that this is a single protein (ie SOS1)?

Reviewer #2 (Remarks to the Author)

The manuscript describes studies of Ras activation by one of its regulators, the GTP/GDP exchange factor SOS. The mechanism of SOS-mediate activation has been extensively studied

using structural and functional methods, but the role of the C-terminal proline-rich (PR) region of SOS has not clearly been elucidated. Indirect evidence for PR inhibition of SOS activity, which can be relieved through SOS-Grb2 interactions, has been recognized since the 90s from deletion/mutant analyses of the multi-domain SOS. However, direct evidence has been hampered by the inability to purify full-length SOS.

This paper is an important contribution to the understanding of Ras activation, as it clarifies the role of the PR domain on SOS. The single-molecule technology has been reported previously and adapted measure Ras GTP exchange on lipid arrays, using fluorescent nucleotide in the presence of full-length/truncated SOS variants from eukaryotic cell lysates.

Below are comments and concerns regarding the paper:

1. line 205-6, "The enhancement of catalysis by truncation of the PR domain involves allosteric regulation mechanism". Regarding this conclusion, which is critical for the paper, can the authors confirm that assays accounted for huge variations in recombinant protein expression (eg, SOS-hdpc vs. SOS-cat). In terms of diluting the lysate (Methods), was this performed with a buffer? Can inhibitors or modulators in the lysate be ruled out for affecting the assay (i.e., some samples would have much more concentrated lysate)?

2. lines 205-206 and line 209 seem redundant, the ideas here are not clearly expressed and require revisions. From the data (Fig 3), one can say that the C-terminal (PR) region antagonizes GTP/GTP-fluor exchange. However, I'm not sure whether, at this point in the paper, one can conclude it is allosteric regulation. Alternative explanations could be that the PR domain binds to Ras, or there could be folding issues (unrelated to allostery), etc... Allostery is further concluded from expression of the catalytic core (lines 211-217) but I suggest the authors remain equivocal until there is stronger evidence for self-inhibition by an allosteric, conformational switch (later in the paper, with the direct binding data, Fig4 and Fig6 experiments with allosteric site mutant).

3. Fig 1c and wide gaussian activity profiles, can authors suggest why there is significant single-molecule heterogeneity SOS activity, even with the endogenous protein. i.e why are some molecules of SOS may more active than others? For non-specialists, it might be useful to have a brief clarification (eg, post-translational modifications, damage/oxidation during lysis, conformation ensembles of SOS, etc...)

4. Fig 1b/c, controls for intrinsic GTPase activity of Ras - it's a little confusing. Presumably without SOS in the corrals there is little change in fluorescence. However, this is not clearly shown by the authors. It appears Supp Fig 8 is the control, but the calculation is done differently (y-axis). It would be useful for readers to reference the background more easily, in order to understand the significance of Figs 1b/c (general dark corrals that are integrated for the gaussian

profiles).

5. A few other questions: Can a binding affinity between the allosteric SOS site and the PR region be determined *in vitro*? The catalytic core is soluble (approx 500-1050), while separately the PR region can also be expressed solubly (Biophys Chem 2013, 175: p.54). Therefore, the experiment seems quite trivial, using any available technique for protein:protein interactions, and may be a useful parameter to know in trying to understand SOS activation via membrane-dependent Grb2 interactions. Through deletion analyses, perhaps only a short segment of the PR region is required for binding to the regulatory domain, and in principle the complex SOS-cat/PR could be co-crystallized.

Overall, this manuscript is an insightful contribution to SOS regulation of Ras. The paper exploits a novel single-molecule technique to provide mechanistic insight into a long-standing question about Ras activation. However, there are nevertheless a few minor shortcomings as a consequence of the technique itself. For example, the steady state assay for SOS activity does not actually reflect the biological activity in cells, since the GEF activity is measured exclusively with GTP loaded molecules. Also, in the Results section, the authors should refrain from extrapolating data to cellular activity without stronger biological evidence. The 'rare highly processive' corrals in FL-SOS are given special biological significance (lines 184-195, Fig.3), which I feel is highly speculative. It is important to remain cautious, bearing in mind that even the '3-sigma' definition for activity (Fig 3) is hypothetical and is not directly linked to cellular activation. Despite these shortcomings, this is a rigorous and insightful study that shapes how we conceptually view the Ras signaling pathway, and will be of general interest to the scientific community.

minor issues:

- line 75 swap 'the both' to 'both the'
- line 206 - 'The both forms...' to 'Both forms...'
- lines 192-5 - I could not find the annotation explaining the 3-sigma threshold, either in the main text, nor the figure legend. I presume it is the grey shading in (Fig 3 d,e,f)
- the non-Gaussian second population in Fig. 3e - where is this? Can an arrow or other marker point this out? Is it the series of spikes toward the right (increasing activities)? It is confusing to readers if this is referred to as a 'second population'.

Reviewer #3 (Remarks to the Author)

The authors expanded their previously reported single-molecule Ras GEF assay to measuring the activity of wild-type and mutant SOS in crude cell lysates. This is an important methodology advancement as it eliminates the need for protein purification and also allows potential consideration of cellular proteins in the environment. Using this method they interrogated the

role of the PR domain in SOS, and found that this domain exerts an autoinhibitory function that is independent of the previously characterized inhibitory function of the N-terminal domain, and that involves blocking of SOS binding to Ras. The experimental design is logical. The data are of good quality and support the authors' conclusions. This new, mechanistic understanding of SOS regulation is important in light of the oncogenic role of Ras and the fact that there are cancer-associated mutations in the PR domain of the SOS gene.

I have a minor issue with the writing of the manuscript: in several instances in the Results section it seems that the conclusion is given before the experiments and results are described, causing confusion as to whether the concluding statement refers to previous reports or results yet to be described. For example, on page 8, second paragraph: "We observed that the classical regulatory mechanism of the N-terminal domains operate...". See also page 6 second paragraph and page 9 second paragraph.

Reviewer #1 (Remarks to the Author):

The manuscript by Lee et al is an extension of the Groves' laboratory's previous work assessing SOS-dependent Ras activation at the single molecule level. They had previously shown that the N-terminal region of SOS inhibits membrane recruitment and downstream signaling. Significant to this manuscript, the authors show that the PR region of SOS also inhibits SOS membrane recruitment and therefore its activation. Further, they show that this is independent of the PIP2-dependent relief of autoinhibition provided by the N-terminal region of SOS.

Overall, the manuscript is of high quality and does help further our understanding of SOS signaling. However, the "mechanism of SOS PR-domain autoinhibition" is not fully explored. What seems to be missing is experimental evidence showing that the PR region's autoinhibition can be relieved physiologically. Presumably, this is due to Grb2 recruitment to the membrane, as Grb2 placed in solution was insufficient to relieve the autoinhibition. See major comment 1 below.

Comment 1. *The major goal of the manuscript is to show the mechanism of SOS PR-domain autoinhibition, and here the prevailing theory is that this autoinhibition is relieved by PR interactions with Grb2 at the membrane. The authors attempted (Fig. 6C) to enhance Ras/SOS binding by adding increasing amounts of Grb2 in solution, however, this did not relieve autoinhibition...the argument being that this needs to occur in the presence of membrane-specific Grb2 interactions. However, it would really bolster the*

paper and the conclusions if the authors could show this.

The Groves laboratory recently published a manuscript in PNAS where they were able to place phosphorylated LAT in their lipid bilayer, and then used this to show LAT:Grb2:SOS higher-order assemblies. Could the authors use a similar approach where both pLAT and RAS were in the membrane, and then repeat the experiment in Fig. 6C?

Response: Grb2 associates with the PR domain of SOS, both in the cytoplasm and in the context of recruitment to phosphotyrosine residues on activated membrane receptors (*Oncogene* 1995 **11**: p.1107; *Science* 1993 **260**: p.1338). Grb2-mediated recruitment of SOS to activated receptors is the primary mechanism by which SOS becomes activated to activate Ras. (*Nat. Struct. Mol. Biol.* 2016 **23**: p.838; *Cell* 2013 **152**: p.1008). These observations together with our results in Fig. 6c, which affirm that Grb2 alone is insufficient to release autoinhibition (as must be the case for receptor-mediated SOS activation to work in cells anyway), lead to the conclusion that membrane-specific Grb2 interactions are required to relieve autoinhibition of the PR domain. While we agree that direct observations of such effects would be nice, there are several reasons why such experiments are beyond the scope of this paper. First of all, while such an assay is conceptually simple, it is technically very challenging and would involve introducing a significant number of new technologies into this work. Second, since it is already well established that Grb2-mediated membrane recruitment activates SOS, such observations would not add to the fundamental conclusions of this work—which concern autoinhibition by the PR domain.

Comment 2. *In Fig. 4, the authors use a nice, full set of truncation mutants to assess SOS/Ras binding on PS or PIP2 containing membranes, showing that the PR region blocks “most” of SOS recruitment/binding to Ras, and that the N-term region makes Ras binding PIP2-dependent. This experiment is complete, well done, and is the best data in the manuscript.*

Response: We appreciate that the reviewer recognized the importance of these findings.

Comment 3. *In Fig 3 the authors show that removal of the PR region increases “catalytic activation” (or increases the kinetic rate of activation...I think that these are the same) on a PS bilayer, and that this is further increased by removal of the HPD domains. If this assay were instead done in the presence of PIP2, would SOS(Cat) and SOS(HDPC) show similar rates of activation? Furthermore, would a SOS(CatPR) construct show a similar level of allosteric activation to the SOS(HDPC) construct on PS, and would this only show differential activity if the assay were performed with PIP2 present?*

Response: The molecular catalytic rate (nucleotide turnover rate of activated single SOS molecules on the membrane) is similar between full-length SOS and SOS^{HDPC} (Fig. 2d). The increased catalytic activity by removal of the PR domain in Fig. 3 is due to the enhanced kinetic rate of allosteric activation (allosteric Ras binding to SOS) (Fig. 4,6).

As the reviewer suggested, these observations lead to expectation that activity of SOS^{HDPC} and full-length SOS will be more sensitive to PIP_2 than other constructs such as SOS^{Cat} and $\text{SOS}^{\text{CatPR}}$. To address this question, we performed the steady state nucleotide exchange assays in the presence of either PS or PIP_2 (Supplementary Fig. 14). SOS^{HDPC} and $\text{SOS}^{\text{CatPR}}$ show similar catalytic activity on the PS bilayer. However, PIP_2 selectively relieves autoinhibition of N-terminal domains in allosteric binding site, thus resulting in higher activity of SOS^{HDPC} than $\text{SOS}^{\text{CatPR}}$. Full-length SOS also shows enhanced catalytic activity on the PIP_2 bilayer. However, full-length SOS is still partially autoinhibited by the PR domain and shows similar activity to $\text{SOS}^{\text{CatPR}}$. These results are consistent with results in Fig. 4 and the overall conclusions of this paper. We added these steady state nucleotide exchange assay data for all constructs used in this paper in Supplementary Fig. 14.

We included the following, additional text in the manuscript:

Addition of line 279-282. “The observed membrane-dependent binding behaviors show good agreement with SOS activity in steady state nucleotide exchange assays, confirming that the membrane recruitment is a critical regulatory mechanism of SOS (Supplementary Fig. 14).”

Minor comments

1. *The manuscript seems “maximally brief”. As it stands parts are read, as many of the complex assays/mechanisms are not explained in detail, making this hard to read for the molecular biology crowd. It would help if the authors would expand them manuscript to make it more broadly accessible. One example of this is on page 5, the authors refer to “state lifetime” and show data in Supplemental figure 7. It would be helpful if this term were explained somewhere (even if in the Supp 7 fig legend) to aid in understanding the manuscript.*

Response: We have added more detailed explanations of the assays and mechanisms to improve the readers’ understanding. Some examples are below:

- a) Detailed processes to calculate individual activity traces in single-molecule SOS activity assays (Fig. 2c) are provided in Supplementary Fig. 5. Catalytic rate and state lifetime are explained. This supplementary figure is mentioned on line 120 and 475, and the caption of Supplementary Fig. 8 (Supplementary Fig. 7 in the original manuscript).
- b) Detailed processes to calculate activity histogram of steady state nucleotide exchange assays are provided in Supplementary Fig. 9. This supplementary figure is mentioned on line 169 and 491.

2. *In Fig 2D, the authors say that the catalytic rate of endogenous GEF, SOS, and SOS(HPCD) were all “similar”...however looking at the data it seems that there is a shift in the peak/distribution of these three populations. Further, similar changes in their earlier papers (see Science 2014 SOS(cat) vs SOS(HPCD)) were viewed as different. Is*

there some statistical way to analyze whether these are in fact “similar” or not on a population level?

Response: We typically observe a broad distribution of the molecular catalytic rates as shown in Fig. 2D. Although the distributions have different average values, they have large standard deviations and significantly overlap with each other. We added an average value and standard deviation of each distribution in the caption of Fig. 2.

3. On page 4 (bottom), when assessing exchange activity in untransfected cells, the authors say that this activity is “presumably SOS”, and allows for direct analysis of endogenous protein. While there is “unpublished data” from the Roose lab mentioned in their 2007 paper that RasGRP proteins are not expressed in 293T cells, do we know the same is true for RasGRF1/2? Also, would it be better to say that they are analyzing an endogenous activity since we are not positive that this is a single protein (ie SOS1)?

Response: Although RasGRF proteins are expressed in 293 cells (*FEBS* 2005 **272**: p.2304), the activity of RasGRF^{Cdc25} (catalytic unit) is much lower compared to SOS^{Cat} when nucleotide exchange occurs on a membrane (*eLife* 2013 2: p.e00813). Therefore, we consider endogenous SOS as a major source of activity in untransfected cell lysate. But we would like to be careful not to exclude other RasGEFs including RasGRF, which might be activated in cells.

We edited the manuscript as below to clarify this point:

- a) Modification of line 128 “In untransfected cells, a few corrals successfully capture GEFs such as endogenous SOS, enabling direct analysis of these endogenously expressed proteins.”
- b) Addition of line 146 “But other endogenous GEF activity might be detected.”

Reviewer #2 (Remarks to the Author):

The manuscript describes studies of Ras activation by one of its regulators, the GTP/GDP exchange factor SOS. The mechanism of SOS-mediate activation has been extensively studied using structural and functional methods, but the role of the C-terminal proline-rich (PR) region of SOS has not clearly been elucidated. Indirect evidence for PR inhibition of SOS activity, which can be relieved through SOS-Grb2 interactions, has been recognized since the 90s from deletion/mutant analyses of the multi-domain SOS. However, direct evidence has been hampered by the inability to purify full-length SOS.

This paper is an important contribution to the understanding of Ras activation, as it clarifies the role of the PR domain on SOS. The single-molecule technology has been reported previously and adapted measure Ras GTP exchange on lipid arrays, using fluorescent nucleotide in the presence of full-length/truncated SOS variants from eukaryotic cell lysates.

Below are comments and concerns regarding the paper:

Comment 1. *line 205-6, “The enhancement of catalysis by truncation of the PR domain involves allosteric regulation mechanism”. Regarding this conclusion, which is critical for the paper, can the authors confirm that assays accounted for huge variations in recombinant protein expression (eg, SOS-hdpc vs. SOS-cat). In terms of diluting the lysate (Methods), was this performed with a buffer? Can inhibitors or modulators in the lysate be ruled out for affecting the assay (i.e., some samples would have much more concentrated lysate)?*

Response: The samples have different concentrations of lysate given appropriate dilutions with a buffer for activity and binding assays. However, we did not observe that different lysate concentrations significantly affect the binding rate and activity of SOS to Ras. In Fig. 4, the binding rates are linearly scaled with SOS concentration even though lysate concentration changes up to 5 folds. For completeness, we performed additional experiments to test the effects of lysate on exchange activity. 2 nM of SOS^{Cat} solutions were supplemented with various concentrations of untransfected lysate up to ~0.6 mg/mL, which is the maximum lysate concentration used in the steady state nucleotide exchange assays. All solutions exhibit similar exchange activity (major populations fitted with Gaussian distribution). The probability of activated states (non-Gaussian populations) appears to show a slight dependency on background lysate concentration (added as Supplementary Fig. 10). However, this dependency is much weaker compared to the effects of flanking domains on activity and may be simply explained by competition with Ras binding proteins in the lysate. We further compared activity of all constructs (full-length SOS, SOS^{HDPC}, SOS^{CatPR} and SOS^{Cat}) in the presence of the same background lysate. Lysate concentrations were matched to the level of full-length SOS (0.64 mg/ml) by adding untransfected cell lysate. N and C-terminal domain-dependent activity was clearly observed with the same background lysate, confirming that measured variations in activity are due to inhibitory effects of flanking domains. This result was added as Supplementary Fig. 14a.

We modified the manuscript to include the experimental results discussed above:

a) Addition of line 170 “We confirmed that other components of the lysate have minimal effects on exchange activity across a range of concentrations (**Supplementary Fig. 10**)”

Comment 2. *lines 205-206 and line 209 seem redundant, the ideas here are not clearly expressed and require revisions. From the data (Fig 3), one can say that the C-terminal (PR) region antagonizes GTP/GTP-fluor exchange. However, I’m not sure whether, at this point in the paper, one can conclude it is allosteric regulation. Alternative explanations could be that the PR domain binds to Ras, or there could be folding issues (unrelated to allostery), etc... Allostery is further concluded from expression of the catalytic core (lines 211-217) but I suggest the authors remain equivocal until there is stronger evidence for self-inhibition by an allosteric, conformational switch (later in the paper, with the direct binding data, Fig4 and Fig6 experiments with allosteric site mutant).*

Response: The purpose of lines 205-209 is to provide the preliminary suggestion that change of SOS activity by truncation of the PR domain involves allosteric effects. The allosteric effects were observed as the enhanced activity of SOS^{HDPC} by truncating the PR domain depends on the nucleotide state of Ras—higher activity on Ras·GTP than Ras·GDP (Supplementary Fig. 11). The nucleotide state sensitivity is the hallmark of allosteric regulation of SOS. This observation leads us to measure single molecule binding assay of SOS in Fig. 4 and Fig. 6. As the reviewer suggested, we revised the manuscript to clarify this point and avoid a premature conclusion until showing the direct binding data in Fig. 4 and Fig. 6.

The modifications of the manuscript are as below:

- a) Lines 205-209 in the original manuscript were revised and shown in lines 212-216. “The enhancement of catalysis by truncation of the PR domain is consistent with allosteric effects. Both forms of enhanced catalytic activity (weak nonspecific activity and processive activity) in SOS^{HDPC} were diminished with Ras·GDP (**Supplementary Fig. 11**). This nucleotide state sensitivity is the hallmark of allosteric regulation of SOS—the allosteric site has a higher affinity to Ras·GTP than Ras·GDP³¹.”
- b) Minor modifications of lines 175, 224, 229 and 233 were made to remove “allosteric”.

Comments 3. Fig 1c and wide gaussian activity profiles, can authors suggest why there is significant single-molecule heterogeneity SOS activity, even with the endogenous protein. i.e why are some molecules of SOS may more active than others? For non-specialists, it might be useful to have a brief clarification (eg, post-translational modifications, damage/oxidation during lysis, conformation ensembles of SOS, etc...)

Response: We observed that SOS molecules fluctuate between distinct multiple activity states on the minutes scale and give rise to broad activity profiles. This dynamic heterogeneity is also observed for SOS molecules prepared by bacteria cell expression (*Science* 2014 345: p.50). The heterogeneity might come from conformational fluctuation and its ensembles. However, little is known about this since a SOS structure in an active conformation on the membrane has not yet been reported.

Comments 4. Fig 1b/c, controls for intrinsic GTPase activity of Ras - it’s a little confusing. Presumably without SOS in the corrals there is little change in fluorescence. However, this is not clearly shown by the authors. It appears Supp Fig 8 is the control, but the calculation is done differently (y-axis). It would be useful for readers to reference the background more easily, in order to understand the significance of Figs 1b/c (general dark corrals that are integrated for the gaussian profiles).

Response: We assume that the reviewer discusses Fig. 2b,c instead of Fig. 1b,c in this comment. In single-molecule SOS activity assays (Fig. 2), the corrals without activated SOS (inactive corrals) exhibit basal fluorescence decrease arising from intrinsic Ras turnover and photobleaching. All fluorescence intensity traces were divided by an average trace of inactive corrals to correct intrinsic Ras turnover and photobleaching factors (described on lines 467-470 in methods). After correction of basal fluorescence

reduction, inactive corrals show constant fluorescence signals. We added a fluorescence intensity trace of inactive corrals in Fig 2c.

Supplementary Fig. 8 in the original manuscript is intrinsic Ras turnover and photobleaching curves for steady state nucleotide exchange assays (Fig. 3) (described on lines 167-170 in the revised main text and 488-492 in methods, as Supplementary Fig. 9).

In order to clarify these points and improve reader's understanding of assays/data analysis, we modified the manuscript as below:

- a) A fluorescence intensity trace of inactive corral was added in Fig. 2c.
- b) Detailed processes to calculate individual activity traces in single-molecule SOS activity assays (Fig. 2c) are provided in Supplementary Fig. 5. This supplementary figure is mentioned on line 120 and 475.
- c) Detailed processes to calculate activity histogram of steady state nucleotide exchange assays (Fig. 3) are provided in Supplementary Fig. 9. This supplementary figure is mentioned on line 169 and 491.

Comments 5. *A few other questions: Can a binding affinity between the allosteric SOS site and the PR region be determined in vitro? The catalytic core is soluble (approx 500-1050), while separately the PR region can also be expressed solubly (Biophys Chem 2013, 175: p.54). Therefore, the experiment seems quite trivial, using any available technique for protein:proteins interactions, and may be a useful parameter to know in trying to understand SOS activation via membrane-dependent Grb2 interactions. Through deletion analyses, perhaps only a short segment of the PR region is required for binding to the regulatory domain, and in principle the complex SOS-cat/PR could be co-crystallized.*

Response: In principle, we agree that characterization of interactions between the PR domain and the catalytic core could help understand SOS activation on membranes if a direct, and strong binding was observed. However, since the catalytic core and the PR domain are physically linked in the full-length protein, characterization of interactions with separate domains may not reveal relevant parameters. Furthermore, functional autoinhibition of the PR domain may not require specific interactions with the catalytic core. When the PR domain is tethered to the catalytic core, a large volume of disordered polypeptide structure would sterically block the allosteric binding site. The co-crystallization experiment is interesting since it may specify the interacting residues. Though well beyond the scope of this paper, this suggestion has been passed to relevant crystallography groups with hopes that it may be addressed at some point.

Overall, this manuscript is an insightful contribution to SOS regulation of Ras. The paper exploits a novel single-molecule technique to provide mechanistic insight into a long-standing question about Ras activation. However, there are nevertheless a few minor shortcomings as a consequence of the technique itself. For example, the steady state assay for SOS activity does not actually reflect the biological activity in cells, since the GEF activity is measured exclusively with GTP loaded molecules. Also, in the Results section, the authors should refrain from extrapolating data to cellular activity without

stronger biological evidence. The ‘rare highly processive’ corrals in FL-SOS are given special biological significance (lines 184-195, Fig.3), which I feel is highly speculative. It is important to remain cautious, bearing in mind that even the ‘3-sigma’ definition for activity (Fig 3) is hypothetical and is not directly linked to cellular activation. Despite these shortcomings, this is a rigorous and insightful study that shapes how we conceptually view the Ras signaling pathway, and will be of general interest to the scientific community.

Response: We revised the lines 184-195 in the original manuscript to clearly convey an idea that the activation threshold ($\mu + 3\sigma$) is hypothetical and different from the actual threshold for cellular activation. The revised texts are shown on lines 186-202.

The detailed modifications are as below:

- a) Modification of line 186. “We propose that these highly processive, long-lived SOS molecules may be disproportionately significant in the context of cell signaling.”
- b) Modification of line 194. “In this context, we analyzed the effects of N and C-terminal regulatory domains by examining a hypothetical activation threshold of the mean plus three standard deviations ($\mu + 3\sigma$) of the major, minimally active population for full-length SOS (**Fig. 3d**).”
- c) Addition of line 200. “It is worth noting that this hypothetical threshold could be different from the cellular activation threshold, which involves more complex signaling networks.”

Other minor issues:

- line 75 swap ‘the both’ to ‘both the’

Response: Changed accordingly (line 75).

- line 206 - ‘The both forms...’ to ‘Both forms...’

Response: Changed accordingly (line 213).

- lines 192-5 - I could not find the annotation explaining the 3-sigma threshold, either in the main text, nor the figure legend. I presume it is the grey shading in (Fig 3 d,e,f) Add 3-sigma annotation in figure.

Response: Notation ($\mu + 3\sigma$) was added in Fig. 3.

- the non-Gaussian second population in Fig. 3e - where is this? Can an arrow or other marker point this out? Is it the series of spikes toward the right (increasing activities)? It is confusing to readers if this is referred to as a ‘second population’. Add arrows signs.

Response: Signs for indicating the highly processive, second populations were added in Fig. 3.

Reviewer #3 (Remarks to the Author):

The authors expanded their previously reported single-molecule Ras GEF assay to measuring the activity of wild-type and mutant SOS in crude cell lysates. This is an

important methodology advancement as it eliminates the need for protein purification and also allows potential consideration of cellular proteins in the environment. Using this method they interrogated the role of the PR domain in SOS, and found that this domain exerts an autoinhibitory function that is independent of the previously characterized inhibitory function of the N-terminal domain, and that involves blocking of SOS binding to Ras. The experimental design is logical. The data are of good quality and support the authors' conclusions. This new, mechanistic understanding of SOS regulation is important in light of the oncogenic role of Ras and the fact that there are cancer-associated mutations in the PR domain of the SOS gene.

I have a minor issue with the writing of the manuscript: in several instances in the Results section it seems that the conclusion is given before the experiments and results are described, causing confusion as to whether the concluding statement refers to previous reports or results yet to be described. For example, on page 8, second paragraph: "We observed that the classical regulatory mechanism of the N-terminal domains operate...". See also page 6 second paragraph and page 9 second paragraph.

Response: We consistently provide the overall conclusion followed by the experiments and results. Although the second paragraph on page 8 falls under this structure, it could be confusing because the experimental results do not follow immediately. Therefore, we reorganized the sentences in this paragraph (shown on lines 251-255).

Reviewers' Comments:

Reviewer #1 (Remarks to the Author):

The manuscript by Lee et al is a revision of an earlier submission. This manuscript assesses the role PR region of SOS inhibits membrane recruitment and therefore its activation. Further, they show that this is independent of the PIP2-dependent relief of autoinhibition provided by the N-terminal region of SOS. This is an exceptional manuscript. I accept that "comment 1" is beyond the scope of the manuscript, and the major points of the manuscript are valid without this experiment. The authors have satisfied all of the other critiques put forth by this reviewer. I recommend publication without further revision.

Significant to the revision, the authors performed additional analysis on the role of SOS1-lipid interactions on the steady state nucleotide exchange rate of truncation mutants of SOS. The authors have also expanded explanations of many of the more technical parts of the manuscript.

Reviewer #2 (Remarks to the Author):

The authors present a revised paper on the role of the proline-rich (PR) domain of a GTP/GDP exchange factor (SOS) in the regulation of Ras activity.

The authors have addressed in detail all of the points raised by my initial review. Their revisions have included additional control experiments. They they have improved their interpretations of data from experiments to better match the figures. The authors have also refrained from over-speculating with regard to the biological activity of the PR domain, given the limitations in their experimental setup.

In particular, the following changes were made:

- additional controls to insure lysate does not contribute to GTP exchange activity (suppl Fig 10)
- discussion of allostery progresses much better in the revisions
- added control - inactive corral trace in 2c
- concerning the biological relevance, authors have included the limitations regarding the assays system, which involves GTP exchange only, as opposed to GDP/GTP exchange

Finally, I accept that the biophysical and structure studies that I proposed are outside the scope of this investigation.

Therefore, I have no further objections to the publication of this manuscript.

Reviewer #3 (Remarks to the Author):

The authors have thoroughly addressed previously raised issues.